# Sex Differences in Cardiovascular Prevention in Type 2: Diabetes in a Real-World Practice Database

**DOI:** 10.3390/jcm11082196

**Published:** 2022-04-14

**Authors:** Anna Ramírez-Morros, Josep Franch-Nadal, Jordi Real, Mònica Gratacòs, Didac Mauricio

**Affiliations:** 1DAP-Cat Group, Unitat de Suport a la Recerca de la Catalunya Central, Institut Universitari d’Investigació en Atenció Primària Jordi Gol (IDIAP Jordi Gol), 08272 Sant Fruitós de Bages, Spain; amramirez.cc.ics@gencat.cat; 2Gerència Territorial de la Catalunya Central, Institut Català de la Salut, 08272 Sant Fruitós de Bages, Spain; 3DAP-Cat Group, Unitat de Suport a la Recerca de Barcelona, Institut Universitari d’Investigació en Atenció Primària Jordi Gol (IDIAP Jordi Gol), 08007 Barcelona, Spain; josepfranch@gmail.com (J.F.-N.); jreal@idiapjgol.info (J.R.); monica.gratacos@gmail.com (M.G.); 4Center for Biomedical Research on Diabetes and Associated Metabolic Diseases (CIBERDEM), Instituto de Salud Carlos III, 08907 Barcelona, Spain; 5Department of Endocrinology and Nutrition, Hospital de la Santa Creu i Sant Pau and Sant Pau Biomedical Research Institute (IIB Sant Pau), 08041 Barcelona, Spain; 6Department of Medicine, University of Vic and Central University of Catalonia, 08500 Vic, Spain

**Keywords:** risk factors, cardiovascular diseases, diabetes mellitus, type 2, gender

## Abstract

Women with type 2 diabetes mellitus (T2DM) have a 40% excess risk of cardiovascular diseases (CVD) compared to men due to the interaction between sex and gender factors in the development, risk, and outcomes of the disease. Our aim was to assess differences between women and men with T2DM in the management and degree of control of cardiovascular risk factors (CVRF). This was a matched cross-sectional study including 140,906 T2DM subjects without previous CVD and 39,186 T2DM subjects with prior CVD obtained from the System for the Development of Research in Primary Care (SIDIAP) database. The absolute and relative differences between means or proportions were calculated to assess sex differences. T2DM women without previous CVD showed higher levels of total cholesterol (12.13 mg/dL (0.31 mmol/L); 95% CI = 11.9–12.4) and low-density lipoprotein cholesterol (LDL-c; 5.50 mg/dL (0.14 mmol/L); 95% CI = 5.3–5.7) than men. The recommended LDL-c target was less frequently achieved by women as it was the simultaneous control of different CVRF. In secondary prevention, women showed higher levels of total cholesterol (16.89 mg/dL (0.44 mmol/L); 95% CI = 16.5–17.3), higher levels of LDL-c (8.42 mg/dL (0.22 mmol/L); 95% CI = 8.1–8.8), and higher levels of triglycerides (11.34 mg/dL (0.13 mmol/L); 95% CI = 10.3–12.4) despite similar rates of statin prescription. Recommended targets were less often achieved by women, especially LDL-c < 100 mg/dL (2.59 mmol/L). The composite control was 22% less frequent in women than men. In conclusion, there were substantial sex differences in CVRF management of people with diabetes, with women less likely than men to be on LDL-c target, mainly those in secondary prevention. This could be related to the treatment gap between genders.

## 1. Introduction

According to the International Diabetes Federation (IDF), the global age-standardized prevalence of diabetes in subjects 20–79 years in 2019 was similar between men and women (9.6% and 9%, respectively) [1]. However, there were more diabetes-associated deaths among women than in men (2.3 vs. 1.9 million) [1].

Large-scale meta-analyses have consistently shown that type 2 diabetes (T2DM) confers a greater excess risk of macrovascular complications in women compared with men. The relative risk of coronary heart disease (CHD) is estimated to be 44% higher in women; the risk of stroke is 27% higher, the occlusive vascular mortality rate is nearly 50% higher, and the risk of vascular dementia is 19% higher [2,3,4]. Regarding microvascular complications, it has been reported that the risk of end-stage renal disease is 38% higher in women than in men [2,3,4]. These disparities have been attributed to the interaction between sex and gender factors in the development, risk, and outcomes of diabetes [3]. Sex differences refer to biology-linked variations, such as sex hormones levels, body composition, and glucose and fat metabolism. Gender differences arise from inequalities in sociocultural processes (e.g., environmental influences, nutritional patterns, lifestyle, or attitudes toward treatment and prevention) [3].

The mechanisms underpinning the biological disparities in the likelihood of developing diabetes-related vascular complications between sexes are not entirely understood. Women develop diabetes at a higher body mass index (BMI) than men, and one of the proposed explanations is that they usually have lower visceral and ectopic fat, which may lead to a slower transition to insulin resistance and diabetes. As a result, women might be exposed longer to hyperglycemia or a suboptimal glucose level state, resulting in greater vascular damage and deterioration of the cardiovascular risk factors (CVRFs) [2,3,4]. In addition to these sex-specific differences, gender dissimilarities in diabetes management and healthcare provision may partially contribute to the diabetes-related increased CVD risk. For instance, although the recommendations on prevention, management, and treatment of diabetes and diabetes-related complications are similar for both sexes, women are less likely than men to receive guideline-recommended care [4]. Indeed, some studies have reported that women are less likely than men to be monitored for foot and eye complications, and they receive less effective management and screening of CV risk factors such as blood pressure (BP), BMI, or smoking status [2,5]. Additionally, the odds of receiving statins, antihypertensive, and antiplatelet medications differ between genders [6,7].

In Spain, a recent observational, prospective study reported that women with T2DM have threefold higher odds of CV death than men [8]. Additionally, previously published cross-sectional and population-based studies indicated a poorest control of CVRF in primary and secondary prevention among Spanish women [9,10,11,12]. In all of these studies, the proportion of women was substantially lower than men, and, most importantly, the baseline characteristics differed significantly between cohorts. For instance, women were on average 2.5–4 years older than men, the duration of T2DM was nearly 1 year longer, they were less likely to smoke, and the prevalence of diabetes-related micro- and macrovascular complications was different between genders. Although these and other differences largely exist in real-life clinical practice, they may limit the interpretation of research findings when traditional cohort matching strategies, stratified analyses, or regression covariate adjustments are used to consider heterogeneity [13]. In contrast, when patients are matched with propensity modeling technologies, the cohorts have a balanced distribution of covariates, thus allowing for equivalent comparisons between groups that can provide inferences about causal effects in observational studies [13].

In Catalonia (Spain), the healthcare system is public and universal. The primary care centers provide first contact and continuing care for persons with any health concerns, and they are usually the principal place where T2DM is diagnosed and managed. The antidiabetic treatment is free of charge for those retired and severely ill people, while active subjects pay just a small part of the cost of the drugs [14]. Briefly, the primary care physicians are responsible for prescribing medications through an electronic prescription that the patient can pick up at the pharmacy. To assess prescribing practices concerning the appropriate use of drugs, the Health Institute of Catalonia uses a quality indicator system created in 2003, the Pharmaceutical Prescription Quality Standard (EQPF) [15]. This study aimed to evaluate whether the pharmacological management of T2DM and the degree of CVRF control in primary care differ between sexes in primary and secondary prevention using a propensity score matching method to balance the inequality of confounding covariates.

## 2. Materials and Methods

### 2.1. Study Design

This was a matched, cross-sectional study including data from patients with T2DM available from the SIDIAP population-based database. This database contains anonymized patient information from the computerized medical records stored in the Electronic Clinical station in Primary Care (eCAP). SIDIAP includes data from about 80% of the Catalonia population (5.835 million subjects) distributed within the 279 primary care centers belonging to the Catalan Health Institute (ICS) [16]. The overall T2DM population has been previously described [17], and this dataset was further used to apply the propensity score method.

The investigation conformed with the principles outlined in the Declaration of Helsinki. The study was approved by the Ethics Committee of the Primary Healthcare University Research Institute (IDIAP) Jordi Gol (P14/018) and registered at ClinicalTrials.gov (NCT04653805).

### 2.2. Study Variables

We used data extracted data from patients aged 31 to 90 years with a diagnosis of T2DM (International Classification of Disease 10 [ICD-10] codes E11 and E14) as at 30 June 2013 who had at least one visit registered with the primary care team in the previous 12 months. For this study, the following variables were used: age, gender, time since diagnosis (years), smoking habit, number of visits with the primary care team in the previous 12 months, estimated glomerular filtration rate (eGFR) with the MDRD (modification of diet in renal disease) formula, presence of diabetic retinopathy (ICD-10 codes E11.3 and H36.0), albumin/creatinine ratio, BMI, glycated hemoglobin (HbA1c), lipid profile (i.e., total cholesterol levels, high-density lipoprotein cholesterol (HDL-c), low-density lipoprotein cholesterol (LDL-c), and triglycerides (TGs)), presence of dyslipidemia (defined as receiving medication for this condition), prescription of glucose-lowering drugs, lipid-lowering drugs (statins or other), blood pressure (BP) (diastolic (dBP) and systolic (sBP)), hypertension (defined as receiving medication for this condition), prescription of hypertension-lowering drugs, and antiplatelet and anticoagulant therapy. Chronic kidney disease was assumed in patients with eGFR < 60 mL/min and/or albumin/creatinine ratio > 300 mg/g. The most recent value registered was used in all cases. For those with a previous CVD, diagnostic codes for macrovascular diseases were collected, including coronary artery disease (CAD; ICD-10 codes I20-I24), cerebrovascular disease (ICD-10 codes I63, I64, G45 or G46), and peripheral artery disease (PAD; ICD-10 code 173.9).

Variables to assess the degree of CVRF control and treatment goals achievement were based on local guidelines [18], i.e., HbA1c ≤ 7% (53 mmol/mol), BP ≤ 140/90 mmHg, and LDL-c < 130 mg/dL (3.37 mmol/L) for primary prevention and <100 mg/dL (2.59 mmol/L) for secondary prevention. Additionally, the same variables were assessed according to the threshold stated by our institution (ICS): HbA1c ≤ 8% (64 mmol/mol), BP ≤ 130/80 mmHg, and LDL-c < 100 mg/dL for primary prevention and LDL-c < 70 mg/dL (1.81 mmol/L) for secondary prevention.

### 2.3. Propensity Score Matching Method

Propensity score matching (PSM) was used to create subpopulations of women and men with T2DM that were balanced in terms of baseline conditions, namely, age, duration of T2DM, number of visits to the primary care team, presence of comorbidities (i.e., hypertension, dyslipidemia, and diabetic retinopathy), eGFR value, albumin/creatinine ratio, and smoking in primary prevention. For the analyses of those in secondary prevention, subjects were also matched for previous macrovascular diseases. Matched groups (male versus female group) were performed (1:1) using the one-to-one nearest neighbor algorithm (with a caliper of 0.1 of the SD of the propensity score on the logit scale) and no replacement. To evaluate PSM quality, we assessed the balance in covariates comparing the absolute difference before and after the matching procedure.

### 2.4. Statistical Analysis

We summarized data as the mean (standard deviation) for continuous variables and number (percentage) for categorical variables by groups. To assess the association between clinical variables and gender, we computed the absolute difference in the means or proportions (Dif) between groups, and we estimated their 95% confidence interval (95% CI). To assess the magnitude of the gender differences, we calculated the relative percentage difference (rDif) between groups. Dif was calculated by subtracting the mean or proportion for women from the mean or proportion for men, and rDif was calculated as the absolute difference divided by the reference value (mean or proportion value of men) multiplied by 100. We performed graphical analyses with smoothing line plots to evaluate whether the potential differences remained over all age ranges. We performed a complete-case analysis excluding missing information for each quantitative variable. All analyses were performed using the R free software environment for statistical computing (v3.5.1) and the “MatchIt” library for the PSM [19].

## 3. Results

A total of 343,969 patients with T2DM were identified in the database. After the matching procedure, there were 70,453 subjects in each primary prevention group and 19,593 in each secondary prevention group (Figure 1). Baseline characteristics in these populations were well balanced (Appendix A).

### 3.1. Primary Prevention

The baseline characteristics of the matched women and men in primary prevention are shown in Table 1. The mean age of the overall population was 66.2 years (SD = 12.2), and the mean duration of diabetes 7.1 years (SD = 5.4) years. Dyslipidemia was present in 52.3% of the patients, hypertension was present in 66.7% of the patients, diabetic retinopathy was present in 6.3% of the patients, and renal impairment was present in 15.9% of the patients.

In this primary prevention population, women had higher BMI than men (Dif = 1.75 kg/m^2^; 95% CI = 1.7 to 1.8) but similar values of HbA1c (Dif = 0.02%; 95% CI = 0.01 to 0.03) and BP (dBP Dif = −0.49 mmHg; 95% CI= −0.5 to −0.4 and sBP Dif = −1.03 mmHg; 95% CI = −1.1 to 0.9). Although the plasmatic TG concentration was comparable between genders, total cholesterol, HDL-c, and LDL-c were higher in women than men (Dif = 12.13 mg/dL, 95% CI = 11.9 to 12.3; Dif = 6.23, 95% CI = 6.1–6.3; Dif = 5.50 mg/dL, 95% CI = 5.3 to 5.7, respectively). Moreover, this sex-difference in total cholesterol and LDL-c was observed across all age ranges (Figure 2A).

Differences by gender in the pharmacological management of T2DM and degree of CVRF control are shown in Table 2 and Figure 3. As for lipid control, statins were more frequently prescribed to women (rDif = 4.7%; Figure 3A). Regarding BP treatment, the prescription of diuretics, beta-blockers, and two antihypertensive drugs was substantially higher in women relative to men (rDif = 16.5%, 10.3%, and 8.1%, respectively). Lastly, women received antiplatelet therapy less often than men (rDif = −15.0%).

The proportion of women who achieved BP target levels was greater in women for both the ≤130/80 mmHg and the ≤140/90 mmHg goals (rDif = 8.8% and 2.5%, respectively). Despite women being more frequently treated with statins than men, fewer women attained the LDL-c ≤ 130 and ≤100 mg/dL thresholds relative to men (rDif = −7.5% and rDif = −14.8%, respectively) (Figure 3B). Regarding glycemic control, the gender differences in the proportion of subjects below the HbA1c ≤ 7 and 8% target was negligible (rDif = −0.7% and 0.1%, respectively). Lastly, the combined achievement of HbA1c, BP, and LDL-c goals was poorest in women relative to men (rDif = −7.4% for LDL-c target < 130 mg/dL and rDif = −14.9% for target ≤ 100 mg/dL).

### 3.2. Secondary Prevention

Baseline characteristics of the matched women and men with T2DM in secondary prevention are shown in Table 3. Overall, subjects were 74.9 years old (SD = 9.9) with a mean diabetes duration of 9.3 years (SD = 6.4). A significant proportion of patients had dyslipidemia (78.8%), and almost all had hypertension (92.6%). Moreover, 11.7% and 35.6% of subjects presented diabetic retinopathy and renal impairment, respectively. Regarding macrovascular diseases, CAD was the most common prior complication (59.9%), followed by cerebrovascular disease (37.6%) and PAD (13.9%).

Similar to what was observed in primary prevention patients, women had higher BMI than men (Dif = 1.69 kg/m^2^, 95% CI = 1.6 to 1.8) but there were no clinically significant differences in HbA1c levels (Dif = 0.11%, 95% CI = 0.09 to 0.1) and BP values (dBP Dif = 0.53 mmHg, 95% CI = 0.4 to 0.6; sBP Dif = 1.00 mmHg, 95% CI = 0.9 to 1.1). Regarding the lipid profile, TG levels in women were comparable to those observed in primary prevention while they were considerably lower in men, which widened the difference between genders (Dif = 11.34 mg/dL; 95% CI = 10.3 to 12.4). All other parameters, such as total cholesterol, HDL-c, and LDL-c were lower than those observed in primary prevention subjects, particularly in men, and all substantially higher among women (Dif = 16.89 mg/dL; 95% CI = 16.5 to 17.3; Dif = 5.77, 95% CI = 5.6 to 5.9; Dif = 8.42 mg/dL; 95% CI = 8.1 to 8.8, respectively). As shown in Figure 2B, these higher total cholesterol and LDL-c levels in women were observed from 40 years onward and persisted in all age groups. In comparison, values in men progressively decreased until around 80 years of age.

Differences by gender in the pharmacological management of T2DM and degree of CVRF control are shown in Table 4 and Figure 4. The proportion of patients prescribed statins was similar between genders (rDif = −0.5%), but treatment with diuretics and three antihypertensive drugs was more frequent in women relative to men (rDif = 18.5% and rDif = 5.3%, respectively) (Figure 4A). Moreover, women received less often antiplatelet and anticoagulant therapy (rDif = −5.7% and −4.0%, respectively). Although the proportion of patients treated with glucose-lowering drugs was similar between groups (rDif = −1%), women were less often prescribed one or more oral antidiabetic drugs (OAD) than men (rDif = −3.8% for one and −18.6% for more than one OAD). Moreover, women were more frequently treated with either insulin alone (rDif = 19.6%) or combined with one or more OAD (rDif =32.2% with one OAD and 8.2% with more than one OAD).

Despite women receiving more intensive antihypertensive treatment, they achieved BP control less frequently than men, either at the ≤130/80 mmHg or at the ≤140/90 mmHg goal (rDif = −6.9% and rDif = −4%, respectively) (Figure 4B). Although the proportion of patients prescribed lipid-lowering treatments was similar between sexes, the targets LDL < 100 mg/dL and <70 mg/dL were less often reached among women (rDif = −16.8% and rDif = −27.7%). Regarding glycemic goals, women showed slightly worse control relative to men (rDif = −5.0% for HbA1c ≤ 7% and −3.5% for HbA1c ≤ 8%). In accordance, the combined target goals of glycemia (HbA1c ≤ 7%), blood pressure (BP ≤ 140/90 mmHg), and LDL-c were less frequently achieved by women than men (rDif = −21.6% for LDL-c < 100 mg/dL and rDif = −29.9% for LDL-c < 70 mg/dL).

## 4. Discussion

The results of this propensity score-matched analysis in patients with T2DM showed that both genders exhibited comparable BP and HbA1c levels, but women had higher BMI and a significantly poorer lipid profile than men. Moreover, there were sex disparities in treatment prescription. Women were more frequently above recommended treatment goals, particularly LDL-c, and the worst overall CVRF control was more pronounced in secondary prevention patients.

The disparities in baseline characteristics between groups were observed in both the primary and secondary prevention cohorts and agree with previous observational studies conducted in Spain and other international large cohort studies assessing sex differences in T2DM risk and management [2,9,10,11,12,20]. Indeed, it has been estimated that women have a BMI nearly 2 kg/m^2^ higher than men at T2DM diagnosis despite similar levels of HbA1c [21,22]. This discrepancy has mainly been attributed to the physiological fat distribution in women, which is characterized by more subcutaneous fat mass and less liver and visceral fat, in addition to greater glucose sensitivity compared with men [20]. Thus, women need to gain more weight and accumulate adiposity to establish a diagnosis of diabetes, which extends the prediabetes state with a result of impairment of CVRF [23].

In our study, women had a worse overall lipid profile relative to men, mainly from approximately 40 years onward. These results agree with recent studies reporting that women with T2DM, particularly after menopause, have higher total cholesterol, LDL-c, and HDL-c than men with T2DM [19,24]. This disparity would lead to a more atherogenic lipid and proinflammatory profile in women with T2DM, in turn linked with an increased cardiometabolic risk [25].

Most notably, our findings confirm inequalities between genders in the pharmacological treatment of T2DM and the ability to reach guideline-recommended targets [2]. The unfavorable lipid profile and difficulties in reaching LDL-c levels below treatment goals among T2DM women regardless of statin treatment are well documented [10,12,19,26]. Although the prescription of statins in our study was slightly more frequent in women in primary prevention and used at similar rates in both genders in secondary prevention, a considerably higher proportion of women were not able to reach the corresponding LDL-c targets relative to men in either condition. One explanation for this disparity could be that women are less likely to receive high-intensity statins than men [27,28]. Other factors may interfere, such as an inadequate adherence to statins (estimated to be 10% greater in women than in men [29]), worse tolerance to this drug class, and less likelihood than men to believe that statins are safe or effective [28].

Although there are divergences in the literature, most studies reported no differences between sexes regarding HbA1c control [2]. Our findings show that the degree of glycemic control was similar in both groups in primary prevention, but it was a little worse among women in secondary prevention. However, women were more likely to be prescribed insulin, alone or in combination. A large population-based study conducted in 415,294 Italian patients with T2DM reported that insulin was more frequently used in women than men when off the HbA1c target [26]. Moreover, that study found a wider use of diuretics in women than men and a slightly higher likelihood of reaching the BP target < 130/80 mmHg [26]. We also found that women received more intense antihypertensive treatment, particularly diuretics, but they were more frequently on BP target only in the case of primary prevention. This agrees with a previous study conducted in our population [10], but contrasts two observational studies conducted in the Netherlands, where stratified analyses found no gender differences in the percentage of patients with or without CVD receiving antihypertensive medication and attaining BP control [20,30]. The discrepancy between studies may be more related to sociodemographic factors than sex-specific differences. For instance, one of the Dutch studies found that women with lower educational level had a higher likelihood of receiving antihypertensive medication when systolic BP > 140 mmHg and were at a higher CVD risk than men [20].

As a result of the suboptimal management of individual CVRF among women (particularly LDL-c levels in primary prevention, and LDL-c and BP levels in secondary prevention), the simultaneous attainment of glucose, lipid, and BP recommended goals was considerably less satisfactory among women even when more intensely treated than men. It has been reported that this gap in CV risk burden is due to the existence of additive factors beyond biological dissimilarities, such as lifestyle, cultural and/or socioeconomic factors, and physician biases [2]. For instance, physical activity levels are lower in women with T2DM than their male counterparts [31], and men were more successful in reducing and maintaining weight than women in most studies [32]. Furthermore, there is still a widespread belief among health professionals that CVD is more prevalent in men, leading to underestimation of the problem among women and, consequently, to undertreatment [23]. Moreover, the intensified multifactorial treatment approach, including nonpharmacological (lifestyle recommendations and close monitoring of laboratory and clinical parameters) and pharmacological treatment, have demonstrated a remarkable benefit for reducing the risk of major cardiovascular events (MACEs) and mortality in high-risk diabetic kidney disease [33]

The main strength of this study is the use of real-world data from a large dataset of primary healthcare services in Catalonia that includes urban and rural areas. Moreover, we used a propensity score matching method to homogenize the sample with a satisfactory reduction in absolute differences of potential confounding variables between genders after the matching procedure. Some studies examined the performance of several methods using PSM for the estimation of different measures of association, showing that the PSM approach estimates with less bias than other regression techniques [34,35,36]. However, the findings of this study must be seen in light of some limitations. Firstly, the cross-sectional design did not allow establishing a causal relationship between the variables. Secondly, we had no data on variables known to contribute to the observed sex dimorphism in diabetes risk and outcome, such as psychosocial risk factors (e.g., socioeconomic status, social support, or educational level) or health behavior (e.g., diet, physical activity, alcohol consumption). Thirdly, we had no data on the doses of the prescribed drugs and whether there were any contraindications (allergies, comorbidities, etc.) that could partially explain gender differences in the disease management. Moreover, we could not assess adherence to the prescribed medications, which may have partly contributed to the observed disparity in CVRF control between sexes. Fourthly, it is not known which comes first, the specific laboratory result (total cholesterol, LDL-c, HDL-c, TGs, and HbA1c) or the particular drug prescription. However, this bias would be present in both groups. Moreover, we did not use the CV risk classification from the 2019 ESC/EASD Guidelines (i.e., moderate, high, or very high CV risk) as the data used predated this recommendation, and the applicable stratification at that time was the requirement of primary vs. secondary prevention. A large population-based study conducted on 373,185 type 2 diabetic subjects in Catalonia reported that at least 50% of them were at very high risk of CV events according to ESC/EASD 2019 classification, and approximately 26% presented with previous CVD [37]. This figure is similar to the proportion of subjects with prior CVD that we included in the secondary prevention group in our study (21.8%). However, categorizing and treating patients according to their CV risk as per the new recommendations will probably provide a more comprehensive and tailored T2DM management than if we only consider the primary/secondary approach [38]. Lastly, we cannot discard that the physician’s sex might have somehow influenced the patient’s assessment and care.

## 5. Conclusions

It is essential that, in the process of care, healthcare professionals, from nurses to physicians and researchers, know and consider that CVD is not only a male issue. Inequalities in the management and control of CVRF in women with T2DM may contribute to an increased risk of CVD compared with men. While more research is needed to elucidate the causes of these inequalities, there is a need to implement gender-sensitive strategies to minimize the existing treatment gap. These should include more stringent follow-up implementing an intensified multifactorial treatment approach to achieve optimal risk factor management and educational programs for healthcare professionals and patients to give visibility and cope with gender disparities.

## Figures and Tables

**Figure 1 jcm-11-02196-f001:**
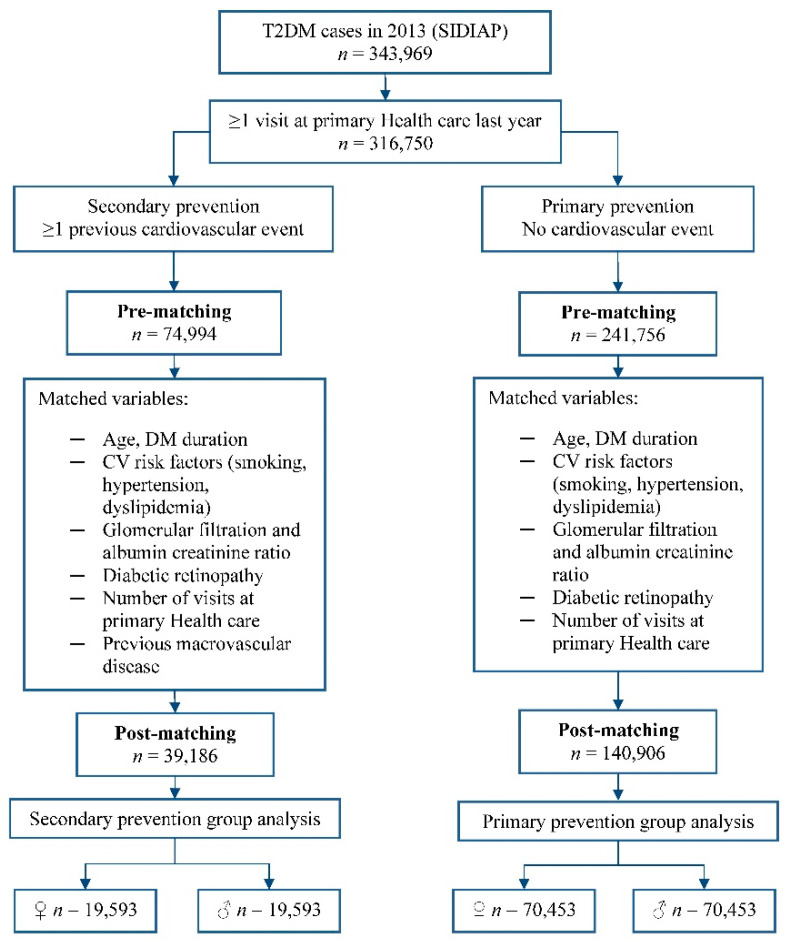
Flow chart of the propensity score matching procedure.

**Figure 2 jcm-11-02196-f002:**
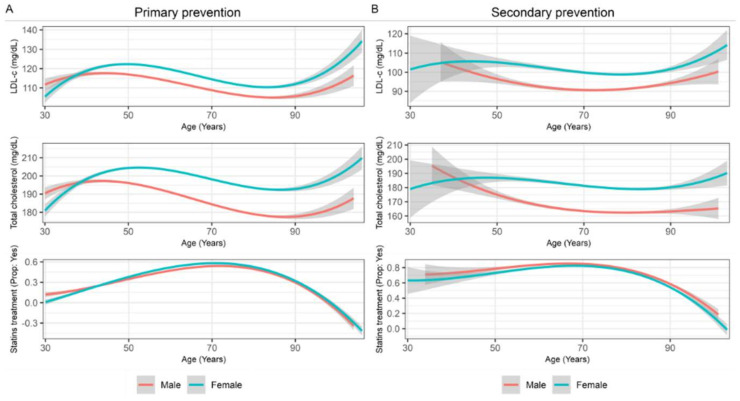
Smoothing line charts with changes in LDL-c, total cholesterol, and statin treatment across age in subjects on primary prevention (**A**) and secondary prevention (**B**) by gender (LDL-c, low-density lipoprotein cholesterol).

**Figure 3 jcm-11-02196-f003:**
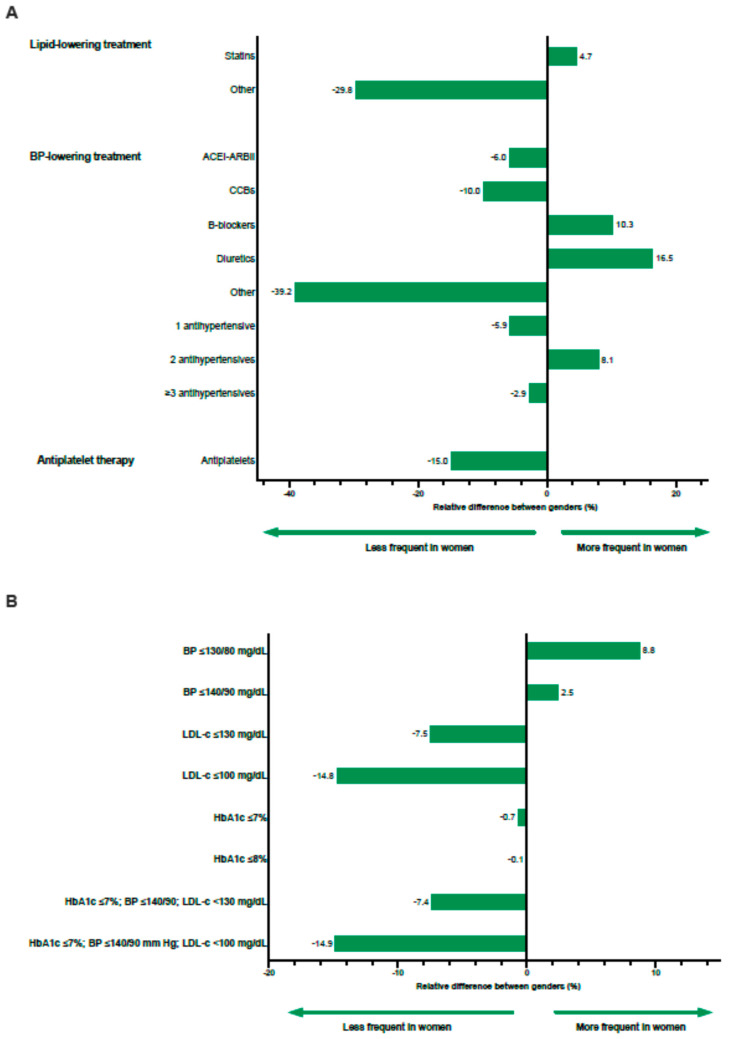
Plot of the relative percent difference between genders for treatments prescribed (**A**) and target achievement (**B**) in the population in primary prevention (ACEI/ARBII, angiotensin-converting enzyme inhibitors/angiotensin II receptor blockers; BP, blood pressure; CCB, calcium channel blockers; HbA1c, glycated hemoglobin; LDL-c, low-density lipoprotein cholesterol; OAD, oral antidiabetic drug).

**Figure 4 jcm-11-02196-f004:**
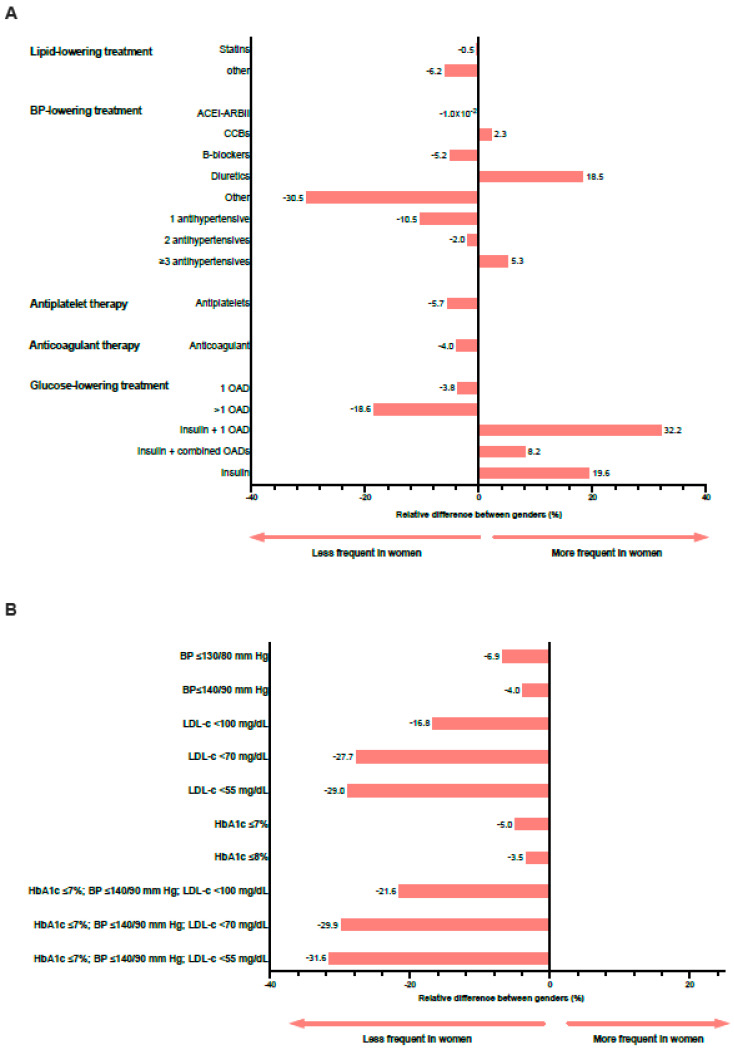
Plot of the relative percentage difference between genders for treatments prescribed (**A**) and target achievement (**B**) in the population in secondary prevention (ACEI/ARBII, angiotensin-converting enzyme inhibitors/angiotensin II receptor blockers; BP, blood pressure; CCB, calcium channel blockers; HbA1c, glycated hemoglobin; LDL-c, low-density lipoprotein cholesterol; OAD, oral antidiabetic drug).

**Table 1 jcm-11-02196-t001:** Baseline characteristics of matched women and men with T2DM in primary prevention by gender.

Variable	*N* Subjects	Women	*N* Subjects	Men	Dif	95% CI	
**Age (years), mean ± SD ***	70,453	66.57 ± 12.22	70,453	65.88 ± 12.20	0.69	0.63	0.75
**Diabetes duration (years), mean ± SD ***		7.10 ± 5.40		7.01 ± 5.34	0.09	0.07	0.12
**Number of visits, mean ± SD *^,†^**		6.39 ± 4.66		6.18 ± 5.08	0.21	0.19	0.24
**Smoking habit, *n* (%) ***	69,001		69,119				
Nonsmoker		51,753 (75.00)		51,118 (73.96)	1.04	0.67	1.42
Smoker		7684 (11.14)		5934 (8.59)	2.55	2.29	2.81
Former smoker		9564 (13.86)		12,067 (17.46)	−3.60	−3.90	−3.30
**BMI (kg/m^2^), mean ± SD**	48,047	31.09 ± 5.80	47,287	29.34 ± 4.47	1.75	1.72	1.78
**HbA1c (%), mean ± SD**	54,055	7.24 ± 1.37	53,476	7.22 ± 1.37	0.02	0.01	0.03
**Dyslipidemia, *n* (%) ***	70,453	37,367 (53.04)	70,453	36,264 (51.47)	1.57	1.13	2.00
**Lipid profile (mg/dL), mean ± SD**							
Total cholesterol	54,561	199.02 ± 37.71	53,976	186.89 ± 37.16	12.13	11.91	12.35
HDL-c	49,918	53.76 ± 13.35	48,986	47.53 ± 12.15	6.23	6.15	6.31
LDL-c		116.10 ± 32.62		110.60 ± 31.29	5.50	5.30	5.70
TGs	51,514	152.23 ± 90.36	50,830	153.70 ± 110.82	−1.47	−2.09	−0.85
**Hypertension, *n* (%) ***	70,453	47,968 (68.09)	70,453	45,949 (65.22)	2.87	2.46	3.27
**Blood Pressure (mmHg), mean ± SD**	59,795		59,067				
dBP		75.95 ± 8.35		76.44 ± 8.62	−0.49	−0.54	−0.44
sBP		133.83 ± 13.25		134.86 ± 12.56	−1.03	−1.10	−0.96
**Diabetic retinopathy, *n* (%) ***	70,453	4418 (6.27)	70,453	4485 (6.37)	−0.10	−0.29	0.10
**Renal disease, *n* (%) *^,$^**	53,782	8617 (16.02)	53,493	8409 (15.72)	0.30	−0.06	0.66

95% CI, 95% confidence interval; BMI, body mass index; dBP, diastolic blood pressure; HbA1c, glycated hemoglobin; HDL-c, high-density lipoprotein cholesterol; LDL-c, low-density lipoprotein cholesterol; Dif, difference between groups; sBP, systolic blood pressure; SD, standard deviation; TGs, triglycerides. * Variables matched between study groups. ^†^ Number of visits with the primary care team in the previous 12 months. ^$^ Renal disease, including eGFR < 60 mL/min and/or albumin/creatinine ratio > 300 mg/g.

**Table 2 jcm-11-02196-t002:** Pharmacological treatment and cardiovascular risk factor control in matched women and men with T2DM in primary prevention by gender.

Variable	*N* Subjects	Women	*N* Subjects	Men	Dif (95% CI)
**Lipid-lowering treatment, *n* (%) ***	37,367		36,264		
Statins		34,933 (93.49)		32,374 (89.27)	4.22 (3.91/4.52)
Other		4653 (12.45)		6430 (17.73)	−5.28 (−5.69/−4.87)
**Antihypertensive treatment, *n* (%) ^†^**	47,968		45,949		
ACEI/ARBII		38,757 (80.80)		39,492 (85.95)	−5.15 (−5.54/−4.76)
CCBs		13,477 (28.10)		14,352 (31.23)	−3.13 (−3.60/−2.68)
Beta-blockers		9893 (20.62)		8586 (18.69)	1.93 (1.53/2.34)
Diuretics		31,429 (65.52)		25,836 (56.23)	9.29 (8.80/9.79)
Other		2570 (5.36)		4054 (8.82)	−3.46 (−3.69/−3.24)
Number of drugs					
1		16,124 (33.61)		16,404 (35.70)	−2.09 (−2.61/−1.57)
2		18,593 (38.76)		16,479 (35.86)	2.90 (2.40/3.40)
≥3		13,251 (27.62)		13,066 (28.44)	−0.82 (−1.27/−0.35)
**Antiplatelet therapy, *n* (%)**	70,453	14,993 (21.28)	70,453	17,645 (25.05)	−3.77 (−4.12/−3.41)
**Target CVRF achievement, *n* (%)**					
BP ≤ 130/80 mmHg	59,795	20,442 (34.19)	59,067	18,558 (31.42)	2.77 (2.32/3.22)
BP ≤ 140/90 mmHg		44,555 (74.51)		42,955 (72.72)	1.79 (1.38/2.21)
LDL-c ≤ 130 mg/dL	49,918	34,707 (69.53)	48,986	36,831 (75.19)	−5.66 (−6.14/−5.18)
LDL-c ≤ 100 mg/dL		16,661 (33.38)		19,213 (39.22)	−5.84 (−6.35/−5.34)
HbA1c, %	54,055		53,476		
≤7		30,262 (55.98)		30,152 (56.38)	−0.40 (−0.91/0.11)
≤8		43,269 (80.05)		42,767 (79.97)	0.08 (−0.32/0.47)
>8		10,786 (19.95)		10,709 (20.03)	−0.08 (−0.47/0.32)
HbA1c ≤ 7%, BP ≤ 140/90 mmHg, LDL-c < 130 mg/dL	43,956	13,173 (29.97)	42,788	13,863 (32.40)	−2.43 (−2.96/−1.90)
HbA1c ≤ 7%, BP ≤ 140/90 mmHg, LDL-c < 100 mg/dL		5935 (13.50)		6787 (15.86)	−2.36 (−2.74/−1.98)

95% CI, 95% confidence interval; ACEI/ARBII, angiotensin-converting enzyme inhibitors/angiotensin II receptor blockers; BP, blood pressure; CCB, calcium channel blockers; CVRF, cardiovascular risk factor; HbA1c, glycated hemoglobin; LDL-c, low-density lipoprotein cholesterol; Dif, difference between groups. * Lipid-lowering treatment, proportion data calculated on the basis of those with dyslipidemia. ^†^ Antihypertensive treatment, proportion data calculated on the basis of those with hypertension.

**Table 3 jcm-11-02196-t003:** Baseline characteristics of matched women and men with T2DM in secondary prevention by gender.

Variable	*N* Subjects	Women	*N* Subjects	Men	Dif	95% CI	
**Age (years), mean ± SD ***	19,593	75.39 ± 9.99	19,593	74.38 ± 9.73	1.01	0.91	1.11
**Diabetes duration (years), mean ± SD ***		9.41 ± 6.50		9.19 ± 6.27	0.22	0.16	0.29
**Number of visits, mean ± SD *^,†^**		8.58 ± 5.99		8.34 ± 6.45	0.24	0.03	0.30
**Smoking habit, *n* (%) ***	19,324		19,342				
Nonsmoker		16,389 (84.81)		16,230 (83.91)	0.90	0.26	1.54
Smoker		958 (4.96)		740 (3.83)	1.13	0.79	1.48
Former smoker		1977 (10.23)		2372 (12.26)	−2.03	−2.57	−1.50
**BMI (kg/m^2^), mean ± SD**	13,022	30.52 ± 5.65	13,181	28.83 ± 4.22	1.69	1.63	1.75
**HbA1c (%), mean ± SD**	14,738	7.31 ± 1.35	14,494	7.20 ± 1.29	0.11	0.10	0.13
**Dyslipidemia, *n* (%) ***	19,593	15,046 (76.79)	19,593	15,814 (80.71)	−3.92	−4.67	−3.17
**Lipid profile (mg/dL), mean ± SD**							
Total cholesterol	15,142	180.71 ± 39.53	14,914	163.82 ± 35.82	16.89	16.46	17.32
HDL-c	13,854	50.87 ± 12.94	13,772	45.10 ± 11.77	5.77	5.62	5.92
LDL-c		100.16 ± 32.80		91.74 ± 29.62	8.42	8.05	8.79
TGs	14,339	152.88 ± 87.11	14,128	141.54 ± 92.12	11.34	10.30	12.38
**Hypertension, *n* (%) ***	19,593	18,113 (92.45)	19,593	18,152 (92.65)	−0.20	−0.64	0.24
**Blood pressure (mmHg), mean ± SD**	17,381		17,326				
dBP		72.15 ± 8.85		71.62 ± 8.76	0.53	0.44	0.62
sBP		134.84 ± 14.51		133.84 ± 13.69	1.00	0.85	1.15
**Diabetic retinopathy, n (%) ***	19,593	2330 (11.89)	19,593	2254 (11.50)	0.39	−0.16	0.95
**Renal disease, *n* (%) *^,$^**	15,067	5456 (36.21)	14,969	5223 (34.89)	1.32	0.255	2.384
**Macrovascular disease, *n* (%) ***	19,593		19,593				
CAD		11,512 (58.76)		11,942 (60.95)	−2.19	−3.12	−1.27
Cerebrovascular disease		7532 (38.44)		7199 (36.74)	1.70	0.79	2.61
PAD		3512 (17.92)		3881 (19.81)	−1.89	−2.58	−1.19
≥2 macrovascular complications		2771 (14.14)		3157 (16.11)	−1.97	−2.53	−1.35

95% CI, 95% confidence interval; BMI, body mass index; CAD, coronary artery disease; dBP, diastolic blood pressure; Dif, difference of means between groups; HbA1c, glycated hemoglobin; HDL-c, high-density lipoprotein cholesterol; LDL-c, low-density lipoprotein cholesterol; PAD, peripheral artery disease; rDif, relative percentage difference between sexes; sBP, systolic blood pressure; SD, standard deviation; TGs, triglycerides. * Variables matched between study groups. ^†^ Number of visits with the primary care team in the previous 12 months. ^$^ Renal disease, including eGFR < 60 mL/min and/or albumin/creatinine ratio > 300 mg/g.

**Table 4 jcm-11-02196-t004:** Pharmacological treatment and cardiovascular risk factor control of matched women and men with T2DM in secondary prevention by gender.

Variable	*N* Subjects	Women	*N* Subjects	Men	Dif (95% CI)
**Lipid-lowering treatment, *n* (%) ***	15,046		15,814		
Statins		14,592 (96.98)		15,407 (97.43)	−0.45 (−0.75/−0.14)
Other		1952 (12.97)		2185 (13.82)	−0.85 (−1.53/−0.16)
**Antihypertensive treatment, *n* (%) ^†^**	18,113		18,152		
ACEI/ARBII		14,549 (80.32)		14,581 (80.33)	−0.01 (−0.75/0.75)
CCB		7584 (41.87)		7427 (40.92)	0.95 (−0.03/1.93)
Betablockers		9314 (51.42)		9850 (54.26)	−2.84 (−3.84/−1.85)
Diuretics		12,805 (70.70)		10,831 (59.67)	11.03 (10.10/11.95)
Other		1988 (10.98)		2867 (15.79)	−4.81 (−5.42/−4.22)
Number of drugs					
1		3037 (16.77)		3400 (18.73)	−1.96 (−2.67/−1.26)
2		5652 (31.20)		5779 (31.84)	−0.64 (−1.54/0.28)
≥3		9424 (52.03)		8973 (49.43)	2.60 (1.60/3.59)
**Antiplatelet therapy, *n* (%)**	19,593	15,203 (77.59)	19,593	16,127 (82.31)	−4.72 (−5.45/−3.99)
**Anticoagulant therapy, *n* (%)**	19,593	2895 (14.78)	19,593	3018 (15.40)	−0.62 (−1.23/0.03)
**Diabetes treatment, *n* (%)**	16,896		17,066		
OAD					
1		6220 (36.81)		6528 (38.25)	−1.44 (−2.44/−0.44)
>1		4093 (24.22)		5076 (29.74)	−5.52 (−6.41/−4.63)
Insulin and 1 OAD		2893 (17.12)		2210 (12.95)	4.17 (3.48/4.87)
Insulin and combined OAD		1532 (9.07)		1430 (8.38)	0.69 (0.17/1.21)
Insulin		2158 (12.77)		1822 (10.68)	2.09 (1.48/2.71)
**Target CVRF achievement, *n* (%)**					
BP ≤ 130/80 mmHg	17,381	6228 (35.83)	17,326	6666 (38.47)	−2.64 (−3.62/−1.66)
BP ≤ 140/90 mmHg		12,372 (71.18)		12,840 (74.11)	−2.93 (−3.82/−2.04)
LDL-c < 100 mg/dL	13,854	7669 (55.36)	13,772	9161 (66.52)	−11.16 (−12.31/−10.02)
LDL-c < 70 mg/dL		2245 (16.20)		3087 (22.42)	−6.22 (−7.07/−5.35)
LDL-c < 55 mg/dL		762 (5.50)		1068 (7.75)	−2.25 (−2.74/−1.77)
HbA1c, %	14,738		14,494		
≤7		7634 (51.80)		7905 (54.54)	−2.74 (−3.88/−1.60)
≤8		11,391 (77.29)		11,607 (80.08)	−2.79 (−3.68/−1.90)
>8		3347 (22.71)		2887 (19.92)	2.79 (1.90/3.68)
HbA1c ≤ 7%, BP ≤ 140/90 mmHg, LDL-c < 100 mg/dL	12,365	2593 (20.97)	12,239	3275 (26.76)	−5.79 (−6.82/−4.76)
HbA1c ≤ 7%, BP ≤ 140/90 mmHg, LDL-c < 70 mg/dL		767 (6.20)		1083 (8.85)	−2.65 (−3.20/−2.09)
HbA1c ≤ 7%, BP ≤ 140/90 mmHg, LDL-c < 55 mg/dL		262 (2.12)		379 (3.10)	−0.98 (−1.28/−0.67)
**Statin treatment and LDL cholesterol target, *n* (%)**	13,854		13,772		
LDL-c < 100 mg/dL and statins		6500 (46.92)		7897 (57.34)	−10.42 (−11.60/−9.24)
LDL-c < 100 mg/dL and no statins		1014 (7.32)		1090 (7.91)	−0.59 (−1.13/−0.06)
LDL-c ≥ 100 mg/dL and statins		4223 (30.48)		3168 (23.00)	7.48 (6.46/8.50)
LDL-c ≥ 100 mg/dL and no statins		2117 (15.28)		1617 (11.74)	3.54 (2.79/4.29)

95% CI, 95% confidence interval; ACEI/ARBII, angiotensin-converting enzyme inhibitors/angiotensin II receptor blockers; BP, blood pressure; CCB, calcium channel blockers; CVRF, cardiovascular risk factor; Dif, difference between groups; HbA1c, glycated hemoglobin; LDL-c, low-density lipoprotein cholesterol; OAD, oral antidiabetic drug; rDif, relative percentage difference between sexes. * Lipid-lowering treatment, proportion data calculated on the basis of those with dyslipidemia. ^†^ Antihypertensive treatment, proportion data calculated on the basis of those with hypertension.

## Data Availability

The data presented in this study are available from the corresponding author upon reasonable request.

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
