# Peer review of "Sex Differences in Cardiovascular Prevention in Type 2: Diabetes in a Real-World Practice Database"

_jcm, 2022, doi:10.3390/jcm11082196_

Round 1
Reviewer 1 Report
The paper is interesting and well written. Methodology is correct. The conclusions are supported by the results.
However, this reviewer raises some issues that need to be addressed by the authors.
1- In the study, the authors distinguished diabetic subjects between those in primary or secondary CV prevention. Actually, this distinction, based on the ESC/EASD guidelines of 2019, is outdated. Therefore, the diabetic population must be divided between subjects with moderate, high, or very high CV risk A diabetic may be in primary prevention but have a very high risk as a diabetic in secondary prevention. This important issue should be addressed in the discussion and added to the limitations of the study.
2- Personalized therapy certainly represents an upgrade in the management of diabetes therapy in both genders. Very recently, in study NID-2 (Cardiovasc Diabetol (2021) 20: 145. Doi: 10.1186 / s12933-021-01343 -1) mortality and MACEs were studied in a type 2 diabetic population in the primary prevention of CV, but at very high CV risk. This randomized multicenter study in which the two genders were equally represented originally demonstrated the ability of a multifactorial therapeutic approach to improve MACEs and overall mortality with short intervention, as well as a long duration of CV protection. Therefore, comprehensive therapy aimed at targeting all major CV risk factors in order to improve mortality and MACEs should be equally mandatory in both women and men. This important issue and the above reference need to be adequately commented on in the discussion and conclusions.
3- According to the above, it would be interesting if the authors define the number of diabetic subjects who reach not only one, but two or more targets.
4- Table 4 shows the percentages of subjects achieving LDL-c target <100 and <70 mg/dL. In reality, the target in diabetics with very high CV risk (and therefore also in secondary CV prevention) is <50 mg/dl. Therefore, authors should indicate the subjects who reach this target.
Author Response
We appreciate the input given by the Reviewers, which enabled us to improve the quality of our Manuscript greatly. In the following pages, we enclose our point-by-point responses to the Reviewers’ comments. Please, note that in the revised version of the Manuscript, changes in response to the Reviewers’ comments are marked up with the “Track changes” function so that they can be easily traced.
Reviewer #1
1- In the study, the authors distinguished diabetic subjects between those in primary or secondary CV prevention. Actually, this distinction, based on the ESC/EASD guidelines of 2019, is outdated. Therefore, the diabetic population must be divided between subjects with moderate, high, or very high CV risk. A diabetic may be in primary prevention but have a very high risk as a diabetic in secondary prevention. This important issue should be addressed in the discussion and added to the limitations of the study.
We agree with the Reviewer that, following the ESC/EASD2019 classification of CV risk, all patients with type 2 diabetes (T2DM) should be treated according to their CV risk, which could be considered as a continuum (Garcia-Moll et al. 2021). However, the database used in the present study was based on the T2DM population in 2013, when the guidelines used in the primary care health system classified CV risk as primary and secondary prevention. Our group reported that at least half of subjects with T2DM in Catalonia were at a very high risk of CV events in 2020 (2). Of note, the proportion of T2DM subjects in secondary prevention that we included in the post-matching population was 21.8%, a figure similar to the 26% reported in 2020 for T2DM patients with previous CVD (Cebrián-Cuenca et al. 2022).
We agree that, although the use of primary and secondary prevention was still widely used in 2013, this is a limitation of the study, and the classification needs to change gradually to incorporate the 2019 ESC/EASD recommendations to better categorize the CV risk in T2DM subjects. As suggested by the Reviewer, the Manuscript has been revised to include this as a limitation of the study and discuss its implications for T2DM care (Discussion section; page 16; lines 387-398):
“Moreover, we did not use the CV risk classification from the 2019 ESC/EASD Guidelines (i.e., moderate, high, or very high CV risk) as the data used predated this recommendation, and the applicable stratification at that time was the requirement of primary vs.- secondary prevention. A large population-based study conducted on 373,185 type 2 diabetic subjects in Catalonia reported that at least 50% of them were at very high risk of CV events according to ESC/EASD 2019 classification, and approximately 26% presented with previous CVD [37]. This figure is similar to the proportion of subjects with prior CVD that we included in the secondary prevention group in our study (21.8%). However, categorizing and treating patients according to their CV risk as per the new recommendations will probably provide a more comprehensive and tailored T2DM management that if we only consider the primary/ secondary approach [38].”
References
- Cebrián-Cuenca, A. M.; Mata-Cases, M.; Franch-Nadal, J.; Mauricio, D.; Orozco-Beltrán, D.; Consuegra-Sánchez, L. Half of Patients with Type 2 Diabetes Mellitus Are at Very High Cardiovascular Risk According to the ESC/EASD: Data from a Large Mediterranean Population. Eur. J. Prev. Cardiol. 2022, 28 (18), e32–e34. https://doi.org/10.1093/eurjpc/zwaa073.
- Garcia-Moll, X.; Barrios, V.; Franch-Nadal, J. Moving from the Stratification of Primary and Secondary Prevention of Cardiovascular Risk in Diabetes towards a Continuum of Risk: Need for a New Paradigm. Drugs Context 2021, 10, 1–3. https://doi.org/10.7573/dic.2021-6-3.
2- Personalized therapy certainly represents an upgrade in the management of diabetes therapy in both genders. Very recently, in study NID-2 (Cardiovasc Diabetol (2021) 20: 145. Doi: 10.1186 / s12933-021-01343 -1) mortality and MACEs were studied in a type 2 diabetic population in the primary prevention of CV, but at very high CV risk. This randomized multicenter study in which the two genders were equally represented originally demonstrated the ability of a multifactorial therapeutic approach to improve MACEs and overall mortality with short intervention, as well as a long duration of CV protection. Therefore, comprehensive therapy aimed at targeting all major CV risk factors in order to improve mortality and MACEs should be equally mandatory in both women and men. This important issue and the above reference need to be adequately commented on in the discussion and conclusions.
We thank the Reviewer for his/her suggested changes. The following changes have been made in the new version that we are submitting:
Discussion section; page 16; lines 363-367:
“Moreover, the intensified multifactorial treatment approach, including non-pharmacological (lifestyle recommendations and close monitoring of laboratory and clinical parameters) and pharmacological treatment, have demonstrated a remarkable benefit on the risk of major cardiovascular events (MACEs) and mortality reduction in high-risk diabetic kidney disease [33]”.
Conclusions section; page 17; lines 406-409:
“These should include more stringent follow-up implementing an intensified multifactorial treatment approach to achieve optimal risk factor management and educational programs for healthcare professionals and patients to give visibility and cope with gender disparities”.
3- According to the above, it would be interesting if the authors define the number of diabetic subjects who reach not only one, but two or more targets.
As suggested by the Reviewer, we have checked this information. Actually, the initial version of the manuscript reported the proportion of subjects achieving more than one target in Table 2 for those in primary prevention and Table 4 for those in secondary prevention. In these tables, the information regarding the achievement of more than 2 targets is shown after the rows reporting the achievement of individual CV risk factors, namely blood pressure, and glycemic and lipid levels.
4- Table 4 shows the percentages of subjects achieving LDL-c target <100 and <70 mg/dL. In reality, the target in diabetics with very high CV risk (and therefore also in secondary CV prevention) is <50 mg/dl. Therefore, authors should indicate the subjects who reach this target.
As suggested by the Reviewer, we have indicated in Table 4 the proportion of subjects who reached the target for diabetes with very high CV risk according to the ESC/EASD Guidelines of 2019, which is <55 mg/dl. Besides, this LDL-c target has also been added to the information regarding the achievement of more than 2 goals. This information has also been properly updated in Figure 4.
Reviewer 2 Report
Dear authors,
Previous studies showed that the risk of CVD is different in men and women with DM, including the outcome of the disease. The aim of the present work is to assess the differences between women and men with DM type 2 in the management and degree of control of cardiovascular risk factors. A matched cross-sectional study was performed. In this study, a large number of T2DM patients included in SIDIAP database (140,906) was used.
The methodology and the results are clearly presented and correlated. The manuscript presents interesting results.
However, some clearer specifications should be made:
- Abstract:
- Line 21: Authors should explain the abbreviation (SIDIAP)
- Introduction:
- Authors should describe the prescribing system of DM treatment, from Spain (is there a national program or a protocol?, who are the prescribers?, who insured the DM costs?, what are the indicators?, what is the role of primary care in DM treatment? etc.). You can use as model, the description of the Romanian system (see introduction): https://www.mdpi.com/1660-4601/17/12/4456/htm
- 4 Statistical analysis:
- Authors should provide details regarding the calculation method of Diff and rDif.
- Results:
- In Statistical analysis section authors used Diff and in table 1-4 they use Dif as abbreviation. Authors should check and correct this abbreviation in all the manuscript.
- Table 1: Some differences seem to be in the calculated values (e.g., Diff for Diabetes duration; % for Dyslipidaemia). Authors should recalculate %, Dif and rDif
- Table 2 and Figure 3 included the values for rDif. I suggest deleteing the column rDif (%) from the table and keeping the Figure.
- Table 4 and Figure 4 included the values for rDif. I suggest deleteing the column rDif (%) from table and keeping the Figure.
- Table 2: How was the percentage (%) calculated? Some differences seem to exist. Also, I observed some differences for rDif (e.g. Lypid lowering treatment) and Dif (e.g., Anti-hypertensive treatment, Target CVRF achievement)
- Table 3: How was the percentage (%) calculated? Some differences seem to exist (e.g. Smoking habit in both groups). Also, I observed some differences for Dif (e.g., Hypertension, CAD)
- Table 4: How was the percentage (%) calculated? Some differences seem to exist. Also, I observed some differences for rDif (e.g. Lypid lowering treatment, Anti-hypertensive treatment) and Dif (e.g., Lypid lowering treatment, Anti-hypertensive treatment,)
- Figure 4: some rDif values are different from rDif values presented in Table 4. The authors should check these values.
- Discussion:
- Authors should present the general advantages of the PSM method.
- Lines 321-323: Authors should rephrase this paragraph
Author Response
We appreciate the input given by the Reviewers, which enabled us to improve the quality of our Manuscript greatly. In the following pages, we enclose our point-by-point responses to the Reviewers’ comments. Please, note that in the revised version of the Manuscript, changes in response to the Reviewers’ comments are marked up with the “Track changes” function so that they can be easily traced.
Reviewer #2
Abstract:
1- Line 21: Authors should explain the abbreviation (SIDIAP).
We thank the Reviewer for this suggestion. In the current version, we have expanded the SIDIAP acronym.
Introduction:
2- Authors should describe the prescribing system of DM treatment, from Spain (is there a national program or a protocol?, who are the prescribers?, who insured the DM costs?, what are the indicators?, what is the role of primary care in DM treatment? etc.). You can use as model, the description of the Romanian system (see introduction): https://www.mdpi.com/1660-4601/17/12/4456/htm
We fully appreciate the Reviewer’s advice. In the new version of the Manuscript, the prescribing system of DM treatment from Catalonia (Spain) is described in the Introduction section (Introduction section; page 2; lines 85-93):
“In Catalonia (Spain), the healthcare system is public and universal. The primary care centres provide first contact and continuing care for persons with any health concerns, and they are usually the principal place where T2DM is diagnosed and managed. The antidiabetic treatment is free of charge for those retired and severely ill people, while active subjects pay just a small part of the cost of the drugs [14]. Briefly, the primary care physicians are responsible for prescribing medications through an electronic prescription that the patient can pick up at the pharmacy. To assess prescribing practices concerning the appropriate use of drugs, the Health Institute of Catalonia uses a quality indicator system created in 2003, the Pharmaceutical Prescription Quality Standard (EQPF) [15].”
References
14. Mata-Cases et al. Trends in the Degree of Control and Treatment of Cardiovascular Risk Factors in People With Type 2 Diabetes in a Primary Care Setting in Catalonia During 2007-2018. Front Endocrinol. 2022 Jan10; 12:810757).
15. Institut Català de la Salut. Estàndard de qualitat de prescripció farmacèutica 2021. http://ics.gencat.cat/web/web/.content/documents/assistencia/31052021_EQPF-2021-GLOBAL-i-MFiC-versio-4.pdf (accessed Apr 4, 2022).
Statistical analysis:
3- Authors should provide details regarding the calculation method of Diff and rDif.
We thank the Reviewer’s advice. The Absolute difference (Dif) was calculated by subtracting the mean or proportion for women from the mean or proportion for men.
Relative difference (rDif) was calculated in percentage terms by taking the summary value (mean or proportion) for men as a reference. Thus, the rDif was calculated as the absolute difference divided by the reference value, which in this case corresponded to the mean and multiplied by 100, that is:
(Dif/(mean or proportion) value for men)*100
This explanation has been included in the Statistical analysis section of the new version for better understanding (Statistical analysis section; page 4; lines 157-160):
“Dif was calculated by subtracting the mean or proportion for women from the mean or proportion for men, and rDif as the absolute difference divided by the reference value (mean or proportion value of men) multiplied by 100”.
Results:
4- In Statistical analysis section authors used Diff and in table 1-4 they use Dif as abbreviation. Authors should check and correct this abbreviation in all the Manuscript.
We thank the Reviewer’s appreciation. The abbreviation has been checked and corrected throughout the Manuscript in the new version (Statistical analysis section; page 4; line 155).
5- Table 1: Some differences seem to be in the calculated values (e.g., Diff for Diabetes duration; % for Dyslipidaemia). Authors should recalculate %, Dif and rDif.
We appreciate the Reviewer’s observation. The observed differences are due to the rounding of the numbers. The calculated values have been checked and recalculated in the new version. Moreover, in Table 1, we are now including the specific sample size for each variable to help understand the calculated percentages.
6- Table 2 and Figure 3 included the values for rDif. I suggest deleteing the column rDif (%) from the table and keeping the Figure.
As suggested by the Reviewer, in Table 2, we have deleted the rDif (%) column.
7- Table 4 and Figure 4 included the values for rDif. I suggest deleteing the column rDif (%) from table and keeping the Figure.
As suggested by the Reviewer, in Table 4, we have deleted the rDif (%) column.
8- Table 2: How was the percentage (%) calculated? Some differences seem to exist. Also, I observed some differences for rDif (e.g. Lypid lowering treatment) and Dif (e.g., Anti-hypertensive treatment, Target CVRF achievement).
We thank the Reviewer for the thorough revision. We did not include the N subject variation of every variable in Table 2. We apologize for the missing information. A new column indicating the number of subjects has been added to allow calculating the percentage. As explained in the footnote of Table 2, lipid-lowering treatment was calculated based on those with dyslipidaemia, and the anti-hypertensive treatment was calculated based on those with hypertension. Additionally, the calculated values (Dif and rDif) have been checked and recalculated in the new version.
9- Table 3: How was the percentage (%) calculated? Some differences seem to exist (e.g. Smoking habit in both groups). Also, I observed some differences for Dif (e.g., Hypertension, CAD)
We thank the Reviewer for this comment. As in the previous question, we did not include the N subject variation of every variable in Table 3. We apologize for the missing information. A new column indicating the number of subjects has been added to allow calculating the percentage. The variables and calculated values (Dif) have been checked and recalculated in the new version.
10- Table 4: How was the percentage (%) calculated? Some differences seem to exist. Also, I observed some differences for rDif (e.g. Lypid lowering treatment, Anti-hypertensive treatment) and Dif (e.g., Lypid lowering treatment, Anti-hypertensive treatment,)
We thank the Reviewer again for this comment. The problem is the same as before, as we did not include the N subject variation of every variable in Table 4. We apologize for the missing information. A new column indicating the N subjects has been added in order to allow calculating the percentage. The variables and calculated values (Dif) have been checked and recalculated in the new version. Beware that, as explained in the footnote of Table 4, lipid-lowering treatment was calculated based on those with dyslipidaemia, and the anti-hypertensive treatment was calculated based on those with hypertension.
11- Figure 4: some rDif values are different from rDif values presented in Table 4. The authors should check these values.
We appreciate the Reviewer’s observation. In Figure 4, rDif values were shown with 1 decimal digit instead of 2 decimal digits, as presented in Table 4. The small differences are due to the rounding of the numbers.
Discussion:
12- Authors should present the general advantages of the PSM method.
We appreciate the Reviewer’s suggestion. As explained in the Discussion section (page 16; lines 370-372), we used a propensity score matching method to homogenize the sample with a satisfactory reduction of absolute differences in potential confounding variables between genders after the matching procedure. Several studies have simulated the performance of several methods based on PSM to estimate different measures of association. These results showed that PSM methods estimates with less bias than other regression techniques. As suggested by the Reviewer, this information has been included in the Discussion section of the Manuscript (page 16; lines 372-375):
“Some studies have examined the performance of several methods based on PSM on the estimation of different measures of association showing that the PSM approach estimates with less bias than other regression techniques [34-36].”
References:
- Austin PC. The performance of different propensity score methods for estimating marginal hazard ratios. Stat Med 2013;32(16):2837-2849.
- Austin PC. The performance of different propensity score methods for estimating marginal odds ratios. Stat Med 2007;26(16):3078-3094.
- Austin PC. The performance of different propensity-score methods for estimating relative risks. J Clin Epidemiol 2008;61(6):537-545
13- Lines 321-323: Authors should rephrase this paragraph
Following the Reviewer’s advice, the paragraph “In our case, the degree of glycaemic control was comparable between genders in primary prevention and only slightly worse among women in secondary prevention” has been rephrased to “Our findings show that the degree of glycaemic control was similar in both groups in primary prevention, but it was a little worse among women in secondary prevention” (Discussion section; page 15; lines 333-335).
Round 2
Reviewer 1 Report
No further comments.